materials science

metal-organic frameworks, electrode materials, lithium–ion batteries, lithium–sulfur batteries

**Author for correspondence:**
Ji Ping Zhu
e-mail: jpzhu@hfut.edu.cn

This article has been edited by the Royal Society of Chemistry, including the commissioning, peer review process and editorial aspects up to the point of acceptance.

# The application of metal-organic frameworks in electrode materials for lithium–ion and lithium–sulfur batteries

## Ji Ping Zhu, Xiu Hao Wang and Xiu Xiu Zuo

School of Materials Science and Engineering, Hefei University of Technology, Hefei 230009, People's Republic of China

JPZ, 0000-0003-4215-377X

Metal-organic frameworks (MOFs) have gained increased attention due to their unique features, including tunable pore sizes, controllable structures and a large specific surface area. In addition to their application in gas adsorption and separation, hydrogen storage, optics, magnetism and organic drug carriers, MOFs also can be used in batteries and supercapacitors which have attracted the researcher's attention. Based on recent studies, this review describes the latest developments about MOFs as battery electrode materials which are used in lithium–ion and lithium–sulfur batteries.

## 1. Introduction

In recent years, along with the development of human society, energy crises and environmental issues have become more important due to the use and exhaustion of fossil fuels. As energy storage devices, batteries and other energy storage devices have become the focus of researchers [1–3].

As we know, lithium–ion batteries are extensively used in mobile phones, laptop computers, portable electronic devices, robots, pure electric cars and hybrid electric cars because of their super properties such as high energy density, light weight and low pollution [4,5]. Compared with lithium–ion batteries, supercapacitors and other batteries (including fuel cells, solar cells, etc.) are not widely used and commercialized, and there is no mature technology, although they are also used in automotive and aerospace applications [6]. Similar to lithium–ion batteries, recently developed lithium–sulfur batteries are also broadly used in aerospace systems and electrical equipment [7–9]. Although there are a lot of papers reporting about the researches and the application of electrode materials in batteries, there are still some

**Figure 1.** Application principle of MOFs in lithium–ion and lithium–sulfur batteries.

effects hindering their commercial application, including life and cycle stability [10]. In order to tackle these problems, several scientists have proposed different methods to improve the capacity and cycle stability of batteries. Metal-organic frameworks (MOFs) are attracting more and more attention as one of the best promising materials for batteries. This typical MOF is an organic–inorganic hybrid material formed through self-assembly of a coordination bond by a central metal ion as a fulcrum and an organic ligand as support, originally discovered and defined by the Yaghi team in 1995 [11]. From the initial simple synthesis to the current comprehensive application, the MOF has grown considerably over the years. Comparing with weak bonds such as hydrogen bonds and van der Waals bonds, MOFs have a stronger coordination bond energy (generally 60–350 kJ mol$^{-1}$), which makes MOFs have a certain stability, able to form permanent pores. In addition, the structure and physical and chemical properties of MOFs are highly designable [12]. Since the MOF was defined, researchers have created more than 20 000 MOFs, and this number is still growing. This is because these different centre-metal ions can form a wide variety of MOF compounds with different organic ligands [13–15]. In addition to the rapid development in drug delivery [16], gas storage and separation [17], and catalysis [18], MOFs have also made considerable progress in the field of electrochemistry recently [19–22]. The application of MOFs in lithium–ion and lithium–sulfur batteries is shown in figure 1.

In this review, we summarize the research and application of MOFs in the area of electrochemistry in recent years, and focus the latest research progress and results of MOFs in the area of lithium–ion and lithium–sulfur batteries.

# 2. MOFs as electrode materials for lithium–ion batteries

## 2.1. MOF-based cathode materials

Recyclable lithium–ion batteries have been extensively used in our life, especially in portable electronic devices, but so far have not been able to meet the needs of super high energy and high density of batteries for large electric devices [23]. Lithium–ion batteries are composed of a cathode electrode, electrolyte and an anode electrode. The working principle is mainly the deintercalation-embedding mechanism of lithium ions: during charge, lithium ions (Li$^+$) move from the cathode electrode to the anode electrode, through the electrolyte and separator diaphragm. Meanwhile, energy is stored in the lithium–ion batteries; during discharge, lithium ions move from the anode electrode to the cathode electrode. At the same time, the lithium–ion batteries release the stored energy. In order to get better lithium–ion batteries, research on the components of lithium–ion batteries is continuing. Being the core component of lithium–ion batteries, research works have been carried out on electrode materials by many researchers. MOFs are one of the best candidates of electrode materials for

**Table 1.** Directly use MOFs as the electrode materials for lithium–ion batteries [2]. CC, charge capacity; DC, discharge capacity; RC, reversible capacity (mA h g$^{-1}$); CN, cycle number. Reproduced with permission.

| MOFs | sample | CC/DC | RC/rate | CN | Vs/Li/Li+ | Refs. |
|---|---|---|---|---|---|---|
| **MOF-based anode materials** | | | | | | |
| MOF-177 | MOF-177 | 110/425 | —/50 | 2 | 0.1–1.6 | [25] |
| $Zn_3(HCOO)_6$ | $Zn_3(HCOO)_6$ | 693/1344 | 560/60 | 60 | 0.005–3 | [26] |
| Mn-LCP | Mn(tfbdc)(4,40-bpy(H$_2$O)$_2$ | 610/1807 | 390/50 | 50 | 0.1–3 | [27] |
| $Co_2(OH)_2BDC$ | $Co_2(OH)_2BDC$ | 1385/1005 | 650/50 | 100 | 0.02–3 | [28] |
| Li/Ni-NTC | Li/Ni-NTC | 601/1084 | 480/- | 80 | 0.01–3] | [29] |
| $Co_3[Co(CN)_6]_2$ | $Co_3[Co(CN)_6]_2$ | 294.2/566.2 | 299.1/20 | 40 | 0.01–03 | [30] |
| $Zn(IM)_{1.5}(abIM)_{0.5}$ | $Zn(IM)_{1.5}(abIM)_{0.5}$ | —/— | 190/100 | 200 | 0.01–3 | [31] |
| $[Cu_2(C_8H_4O_4)4]n$ | $[Cu_2(C_8H_4O_4)_4]_n$ | 194/1492 | 161/48 | 50 | 0.01–2.5 | [32] |
| MONFs | Asp-Cu nanofibers | 334/1255 | 233/50 | 200 | 0.01–3 | [33] |
| Ni–Me$_4$bpz | Ni–Me$_4$bpz | —/320 | 120/50 | 100 | 0.01–3 | [34] |
| Mn-BTC | Mn-BTC | 694/400 | 100/100 | 100 | 0.01–2 | [35] |
| **MOF-based cathode materials** | | | | | | |
| MIL-53 | MIL-53(Fe) | 77/80 | 71/0.025 C | 50 | 1.5–3.5 | [36] |
| MIL-53 | MIL-53(Fe) quinone$_1$ | 77.5/93 | 73/0.1 C | 8 | 1.5–3.5 | [37] |
| MIL-68 | MIL-68(Fe) | 31/40 | 32/0.02 C | 12 | 1.5–3.5 | [38] |
| MIL-136 | MIL-136(Ni) | —/— | 10/10 C | 50 | 2–4 | [39] |
| Co-BTC | Co-BTC | 622/1739. | 750/— | 200 | 0.01–3 | [40] |
| MOPOF | $K_{2.5}(VO)_2(HPO_4)_2(C_2O_4)$ | 81/62 | 70/40 | 60 | 2.5–4.6 | [41] |
| MOPOF | $Li_2[(VO)_2(HPO_4)_{1.5}(PO_4)_{0.5}(C_2O_4)]$ | 78/55 | 80/12.5 | 25 | 2.5–4.5 | [42] |
| MIL-101 | MIL-101-Fe | 65 | 30 | | 2–3.5 | [43] |
| Cu(2,7-AQDC) | Cu(2,7-AQDC) | 147 | 105/1 Ma | 50 | 1.7–4 | [44] |
| MOF-74(Ni) | β-NiS | —/— | 300/60 | 100 | 0.5–3 | [45] |
| MIL-47 | VIV(O)(BDC) | —/— | 82/10 | 50 | 1.5–4 | [46] |
| MIL-132 | K2(TTF-TC)H$_2$ | | 50/10 C | | 2.3–3.75 | [47] |
| RbxMnIIy[FeIII(CN)$_6$]·nH$_2$O | | 60/50 | | | 2.0–4.3 | [48] |
| MnIII[MnIII(CN)$_6$] | | 197/30 | | | 2–4.2 | [49] |
| $K_{2.5}[(VO)_2(HPO_4)_{1.5}(PO_4)_{0.5}(C_2O_4)]$ | | 65/40 | | | 2.5–4.6 | [41] |
| [Mn(H$_2$O)][Mn(HCOO)$_{2/3}$ (H$_2$O)$_{2/3}$]$_{3/4}$[Mo(CN)$_8$]·H$_2$O | | | 30/10 | | 3–4.3 | [50] |

lithium–ion batteries because of their high porosity and structural control [24]. Interesting results have been achieved in the past few years. As shown in table 1, we summarize the application of MOFs as electrode materials for lithium–ion batteries directly.

The first report in 2007 noted that MIL-53(Fe) can be directly used as a cathode electrode material by reversibly inserting lithium ions. Tarascon *et al.* [36] synthesized MIL-53(Fe) by a solvothermal process and thermogravimetric analyses. The working principle of the MIL-53(Fe) electrochemical reaction is $Fe^{3+}$ to $Fe^{2+}$ and simultaneously lithium ions are embedded in the structure. As shown in figure 2, about 0.6 lithium ions can be inserted into MIL-53 at a C/40 rate during each Fe reduction process. The electrode discharge capacity reaches 75 mA h g$^{-1}$ under 1.5–3.5 V, although this capacity is not high, it is a milestone improvement. It is of great significance for people to explore MOF as a lithium ion energy storage material. Tarascon *et al.* [37] then explored the increase in electrochemical capacity by adsorption of 1,4-benzoquinone (electro-active molecules); 1,4-benzoquinone molecules theoretically can accept two electrons per molecule and can also act as a redox medium to enhance electron transfer in MIL-53(Fe).

Despite the reaction of pure quinone, theoretically it reacts with two lithium ions and shows a reversible reaction with only 0.5 lithium ions, because the quinone molecules dissolve into the electrolyte. The benefit of quinone absorption is to increase the capacity of MIL-53(Fe) from 75 to

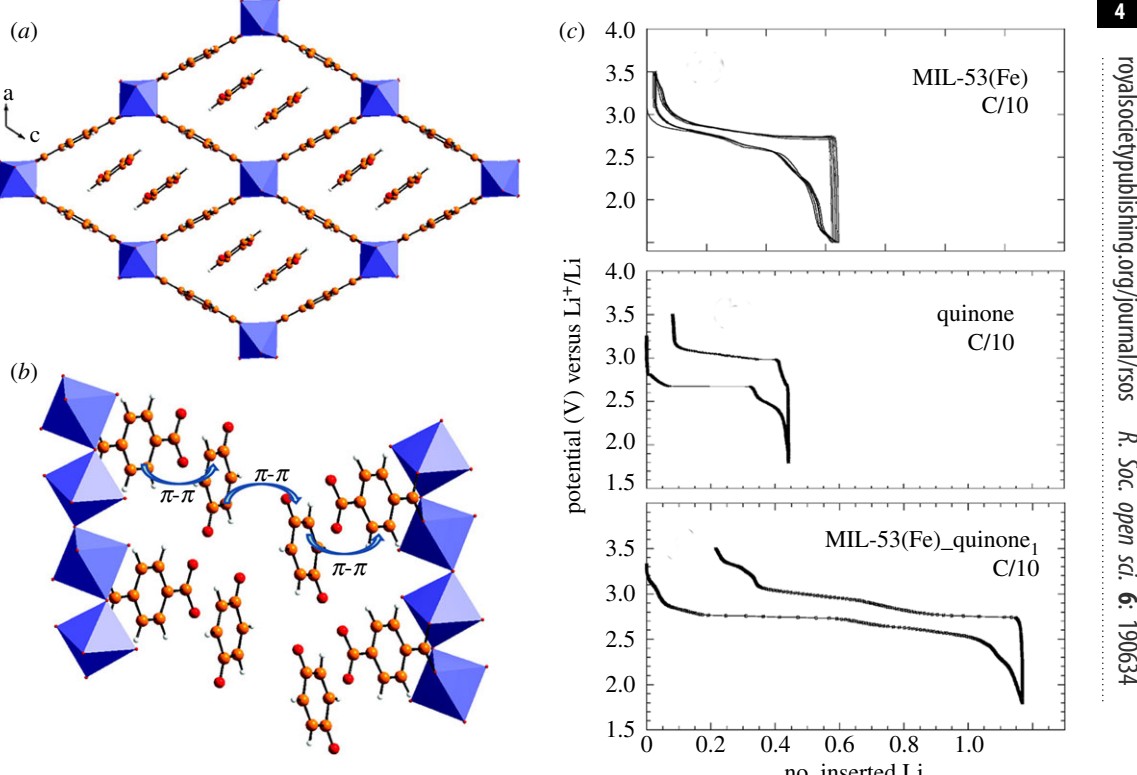

**Figure 2.** Drawing of the unit MIL-53(Fe) quinone$_1$ structure with, in (*a*) and (*b*); (*c*) the relationship between the voltage composition and the number of lithium ions embedded is shown [36]. Images reproduced with permission.

93 mA h g$^{-1}$ under the same conditions. However, this capacity cannot be maintained for too long time. After five cycles, the capacity of MIL-53(Fe)-quinone is rapidly attenuated and the capacity of MIL-53(Fe) without quinone is the same. Electrochemical learning of MIL-53(Fe) provides an idea for the study of MOFs directly as cathode materials.

Although MOFs were directly developed as cathode electrode materials, the battery capacity with MOFs as the cathode materials has never been increased to more than 300 A h g$^{-1}$. Devic *et al.* [38] on exploring the electrochemical performance of MIL-68(Fe) reported that the operating voltage is from 1.5 to 3.5 V, and the Li storage capacity per Fe atom is 0.35 lithium ions, which is lower than MIL-53(Fe), so the electrode capacity of MIL-68(Fe) is only 40 mA h g$^{-1}$. Meng *et al.* [43] demonstrated that each Fe in MIL-101(Fe) can hold 0.62 lithium ions, although the redox chemistry that occurs is not completely reversible, this incomplete reversible redox reaction with the previous MIL-53(Fe) and MIL-68(Fe) is the same; the principle of valence change of iron ions is the same. MIL-101(Fe), MIL-53(Fe) and MIL-68(Fe) all contain Fe as the central metal ion and serve as cathode materials. It shows us the development of MOFs containing Fe directly as the electrode materials.

However, there are also many MOFs that cannot be used as electrode materials because of their poor electrochemical capacity and electrochemical stability. Devic *et al.* [39] reported that MIL-136(Ni,Co), a MOF material, is not suitable to be used as electrode materials because MIL-136 can neither carry out the redox conversion reaction nor has good lithium ion mobility. Unlike MIL-136(Ni,Co), Gu *et al.* [40] synthesized a kind of Co-containing MOFs by the solvothermal method. The Co-based metal-organic framework (Co-MOF) (Co-BTC) was directly used as anode materials without calcination treatment. They exhibited good electrochemical performance. The initial discharge capacity reaches 1739 mA h g$^{-1}$ at 100 mA g$^{-1}$. Through structural analysis and performance testing, they found that some MOFs have linkers that promote effective and reversible charge transfer and stronger π–π interaction between conjugated carboxylates, both of which play a key role in excellent Li storage.

The metal organophosphate open frame (MOPOF) composed of transition metal phosphate is a hybrid material with a multi-dimensional structure. Like other MOFs, MOPOF is known for its structural diversity and industrial applications. Vittal *et al.* [41] taking K$_{2.5}$[(VO)$_2$(HPO$_4$)$_{1.5}$(PO$_4$)$_{0.5}$(C$_2$O$_4$)] (MOPOF) as an example, as shown in figure 3, showed that K ions are deintercalated from the frame during charging and discharge cycles. The Li ion and K ion are intercalated and deintercalated, but as the ion mobility

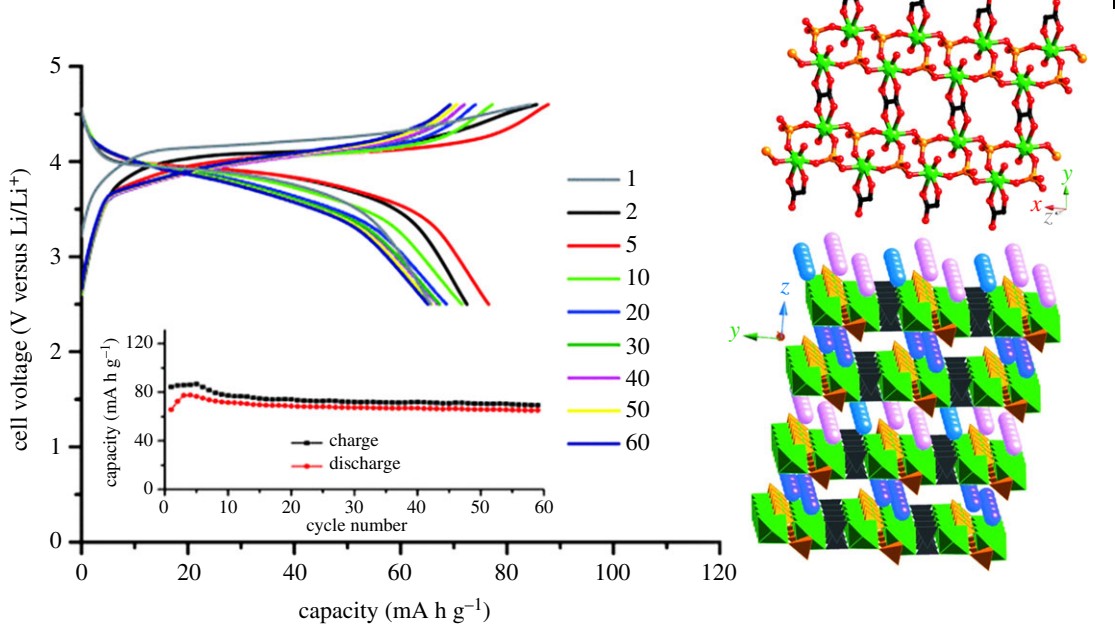

**Figure 3.** Crystal structure diagram and galvanostatic charge–discharge profiles (at cycles given) and cyclability (inset) of dehydrated crystals of $K_{2.5}[(VO)_2(HPO_4)_{1.5}(PO_4)_{0.5}(C_2O_4)]$ [41]. Images reproduced with permission.

increases with respect to the ionic radius, the lithium-rich batteries help lithium ions to migrate and pass through electrodes in charge–discharge cycles instead of K ions which are heavier than lithium ions. At a voltage of 2.5–4.6 V, the cyclic reversible capacities of $K_{2.5}[(VO)_2(HPO_4)_{1.5}(PO_4)_{0.5}(C_2O_4)]$ at the end of 60 cycles at 0.4, 1 and 2 C are 68, 58 and 40 mA h g$^{-1}$, respectively. Another similar MOPOF, $Li_2(VO)_2(HPO_4)_2(C_2O_4)\cdot6H_2O$, is also synthesized by Vittal et al. [42]. At a voltage of 2.5–4.5 V, the electrode with $Li_2(VO)_2(HPO_4)_2(C_2O_4)\cdot6H_2O$ as the electrode material showed a cyclic capacity of 80 mA h g$^{-1}$ when the current ratio is 0.1 C, and the capacity loss rate was low.

The MOF, which also has Cu ion as the central metal ion, is also a promising candidate for cathode electrode materials [44]. Awaga et al.'s Cu-based MOF Cu(2,7-AQDC) (2,7-$H_2$AQDC = 2,7-anthraquinonedicarboxylic acid) synthesized by the solvothermal method has a first discharge capacity of 147 mA h g$^{-1}$ at 1.7–4 V, and then the initial capacity of the battery rapidly decreases in 105 cycles and in the end the capacity is stable to about 105 mA h g$^{-1}$. This MOF electrode material has extremely high battery recyclability, and this MOF with a solid framework is also the first to show independent redox-active sites on central metal ions and organic ligands.

In addition to these lithium–ion battery cathode materials, there are some lithium–ion performance cathode materials that have not been described as having excellent performance, including MOF-74 (Ni) [45], MIL-47 [46], MIL-132 [47], $Rb_xMn^{II}_y[Fe^{III}(CN)_6]\bullet nH_2O$ [48], $Mn^{III}[Mn^{III}(CN)_6]$ [49], $K_{2.5}[(VO)_2(HPO_4)_{1.5}(PO_4)_{0.5}(C_2O_4)]$ [41], $[Mn(H_2O)][Mn(HCOO)_{2/3}(H2O)_{2/3}]_{3/4}[Mo(CN)_8]\bullet H_2O$ [50] and many others lithium–ion battery cathode materials.

## 2.2. MOF-based anode materials

MOF-177, first reported MOFs applied as anode materials by Chen et al. [25], show higher irreversible capacity in the initial discharge, lower stable capacity in the second and subsequent charge–discharge cycles; the MOF-177 (Zn$_4$O(1,3,5-benzenetribenzoate)$_2$) was prepared from Zn(NO$_3$)$_2\bullet$6H$_2$O and H$_3$BTB by the solvothermal reaction. The performance of the battery with MOF-177 as the electrode material is relatively poor. As shown in figure 4, electrochemical tests show that MOF-177 has a reversible capacity of more than 400 mA h g$^{-1}$ in the initial cycle and only 105 mA h g$^{-1}$ during the second charge–discharge cycle. Compared with the initial charge–discharge cycle, the capacity loss in the second charge–discharge cycle is up to three-quarters. It can be concluded that in the initial discharge, an irreversible process leads to large-capacity attenuation. The main reason is that during the first charge and discharge cycles, the solid electrolyte boundary film (SEI) is passivated. The passive film formed can effectively prevent the solvent molecules from passing through. However, Li ions can be embedded and released freely through the passivation layer, so there will be second

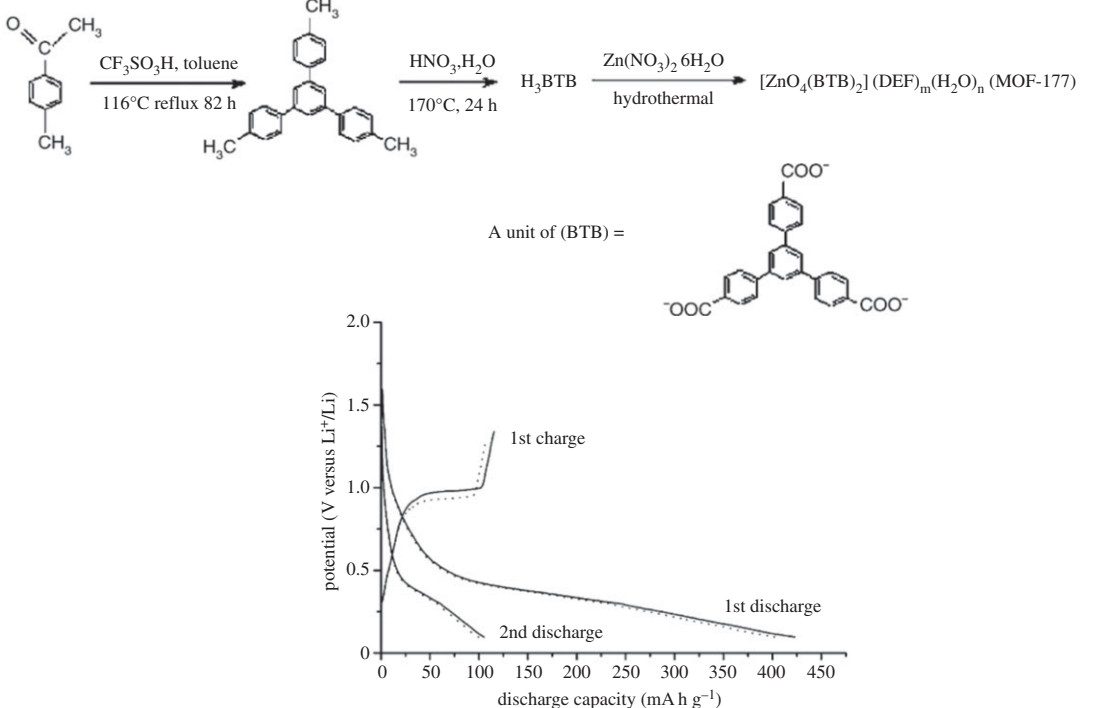

**Figure 4.** Synthesis of MOF-177 and discharge–charge performances of the electrodes made with the as-synthesized MOF-177 microcubes (solid lines) and microcuboids (dotted lines) [25]. Images reproduced with permission.

large-capacity losses in charge–discharge cycles. This second large loss appears in many MOFs directly used as electrode materials. From the point of view of the reaction, the irreversible capacity attenuation is mainly caused by electrochemical processes including irreversible or reversible lithium ion intercalation/deintercalation during electrochemical processes. This reversible or irreversible lithium ion intercalation/deintercalation is quite different from the neutral molecule adsorption/desorption occurring in previously reported MOFs. The adsorption and desorption of neutral molecules have no chemical changes except physical changes. The insertion/deintercalation process of lithium ions involves not only physical changes but also electrochemical reduction/oxidation. Vittal *et al.* [26] reported the working principle of a recyclable Zn–Li system and its application in electrochemistry. Its electrochemical working principle is as follows:

$$Zn_3(HCOO)_6 + 6Li^+ + 6e^- \leftrightarrow 3Zn + 6HCOOLi \tag{2.1}$$

and

$$3Zn + 3Li^+ + 3e^- \leftrightarrow 3LiZn. \tag{2.2}$$

It is different from the Zn-ZnO/ZnLi system, reversible conversion of MOFs in almost all MOFs systems (including $Zn_3(HCOO)_6$) determines whether they can achieve excellent lithium storage performance. The $Zn_3(HCOO)_6$ has a stable capacity of about 560 mA h g$^{-1}$ at 60 mA g$^{-1}$ when the voltage range is 0.005–3 V, up to 60 cycles. The performance experiments show that the metal formate skeleton can react reversibly with lithium through the conversion reaction shown in the chemical equation above. The simple formate hydrochloride MOF has been proved to be a viable new anode material. Wang *et al.* [27] reported that the electrode reaction that may occur in the Mn-LCP electrode is similar to the conversion reaction of the $Zn_3(HCOO)_6$ electrode: [Mn(tfbdc)(4,4'-bpy)(H$_2$O)$_2$] + $2Li^+ + 2e^- \rightarrow Mn + [Li_2(tfbdc)(4,4'-bpy)]$.

As shown in figure 5, the discharge-specific capacity of Mn-LCP is higher than 1800 mA h g$^{-1}$ when the voltage range is 0.1–3 V in the first cycle, but it rapidly reduces to 552 mA h g$^{-1}$ in the second cycle. Compared with the initial cycle, the capacity loss of the second cycle is 69.5%. The main reason is because of the formation of the SEI film, which we have already introduced. The electrochemical capacity of the Mn-LCP still shows 390 mA h g$^{-1}$ stable and reversible lithium storage capacity from the 4th cycle to the 50th cycle, which indicates that the Mn-LCP electrode has good electrochemical cycle stability.

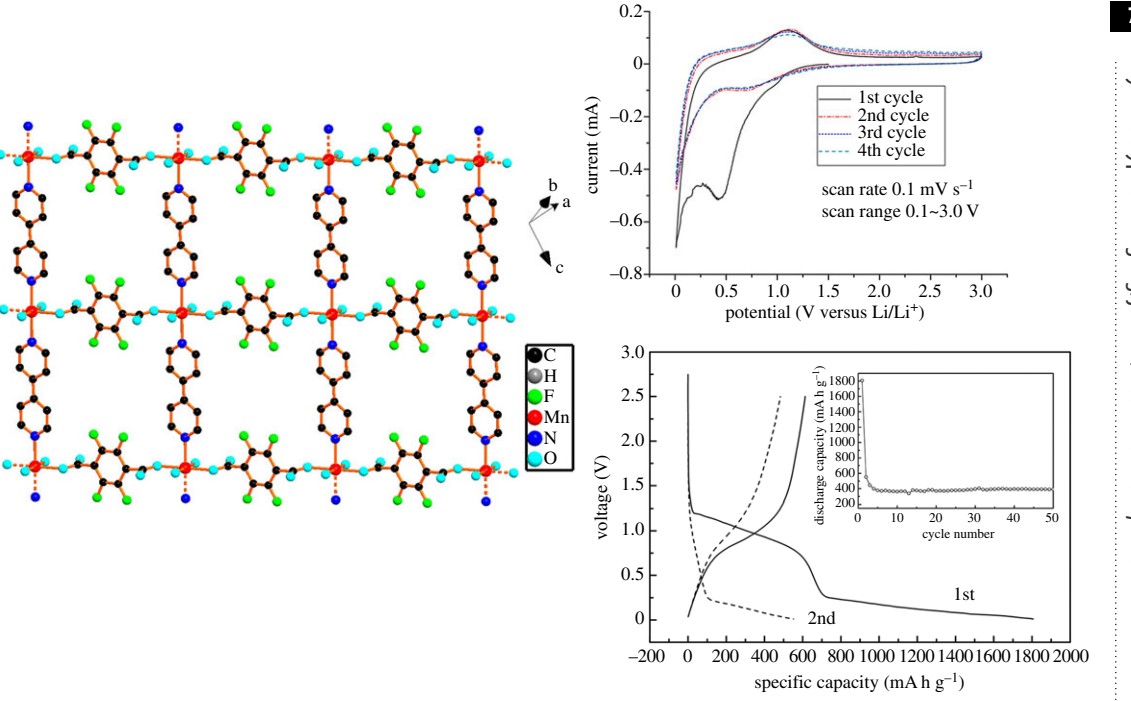

**Figure 5.** Two-dimensional layer structure and electrochemical properties of Mn—LCP [27]. Images reproduced with permission.

Li *et al.* [28] reported a three-dimensional MOF with high crystallinity, $Co_2(OH)_2BDC$, first synthesized by the solvothermal method. It shows excellent cycle stability and extremely large reversible capacity. It has a cyclic capacity of 650 mA h g$^{-1}$ until the 100th cycle when the voltage range is 0.02–3 V and the current density is 50 mA g$^{-1}$, much better than other anode materials containing BDC.

Sun *et al.* [29] reported that metal-1,4,5,8-naphthalenetetracarboxylates (metal = Li and/or Ni) were synthesized and investigated as anode materials. As shown in figure 6, Li-NTC and Ni-NTC and Li/Ni-NTC show excellent electrochemical properties and good electrochemical stability through electrochemical performance tests. The first discharge-specific capacity and charge-specific capacity of Ni-NTC are 1823 and 982 mA h g$^{-1}$ under specific conditions, respectively. After 80 cycles, the specific capacity dropped to 248 and 246 mA h g$^{-1}$; for Li-NTC, the electrode capacity significantly decreased in the initial five cycles. During the initial five cycles, the charge and discharge capacity of Li-NTC decreased to 568 and 554 mA h g$^{-1}$, respectively. After the 74th cycle, the decrease in charge and discharge capacity gradually slowed down. After 80 cycles, the charge and discharge capacities were kept at 468 and 458 mA h g$^{-1}$, respectively. Li/Ni-NTC composite electrodes provided first discharge and charge capacities of 1084 and 601 mA h g$^{-1}$, respectively, and became 482 and 475 mA h g$^{-1}$ after 80 cycles. From the above, it can be concluded that the mixed electrode has higher capacity and better cyclic stability than Li-NTC and Ni-NTC electrode materials. For mixed Li/Ni-NTC electrodes, the Ni-NTC component can increase electron conductivity while the Li-NTC component can improve structural stability. Therefore, Li/Ni-NTC is an anode material with good electrochemical properties and cycle stability. Zhang *et al.* [30] synthesized a typical Prussian blue analogue, $Co_3[Co(CN)_6]_2$, which showed a reversible electrode capacity of 299.1 mA h g$^{-1}$ when the voltage range is 0.01–3 V. It also manifests excellent rate performance; the capacity remains at 34% as the current density increases from 20 to 2000 mA g$^{-1}$. This is also a typical example of using MOFs directly as anode materials.

Chen *et al.* [31] reported that a functionalized MOF with hydrophobic and polar functional groups, BMOF, was synthesized by the hydrothermal method. It has significant thermal and chemical stabilities. Expected BMOF has excellent reversible lithium charge/discharge amount based on DFT calculation. The rate cyclicity of BMOF discharge electrodes at different current densities is shown in figure 7. The charge–discharge capacity of BMOF was stabilized after 10 charge–discharge cycles, and no obvious capacity loss was found during 200 cycles. For BMOF when the discharge current density is 100 mA g$^{-1}$, the capacity even increased to 190 mA h g$^{-1}$ after 200 charge–discharge cycles. Its Coulomb efficiency (ratio of charge-specific capacity to discharge-specific capacity) is close to 97% after 10 charge–discharge cycles. The sample obtains almost 100% Coulomb efficiency during 200

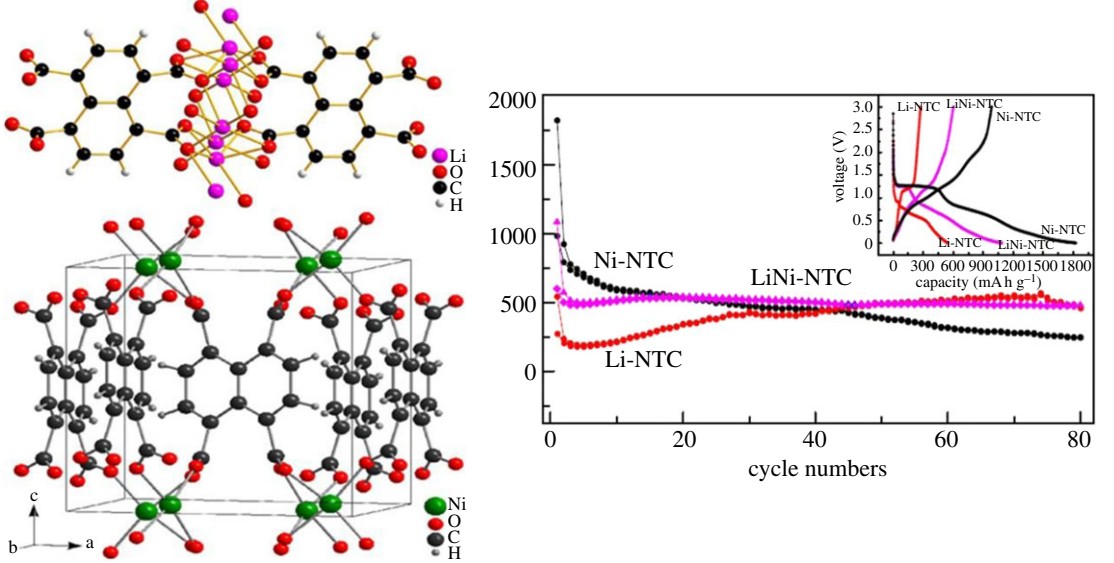

**Figure 6.** The molecular packing diagram of Li-NTC and Ni-NTC in the unit cell and cycling properties of Li-NTC, Ni-NTC and Li/Ni-NTC, the inset is the voltage profile of the samples in the initial cycle [29]. Images reproduced with permission.

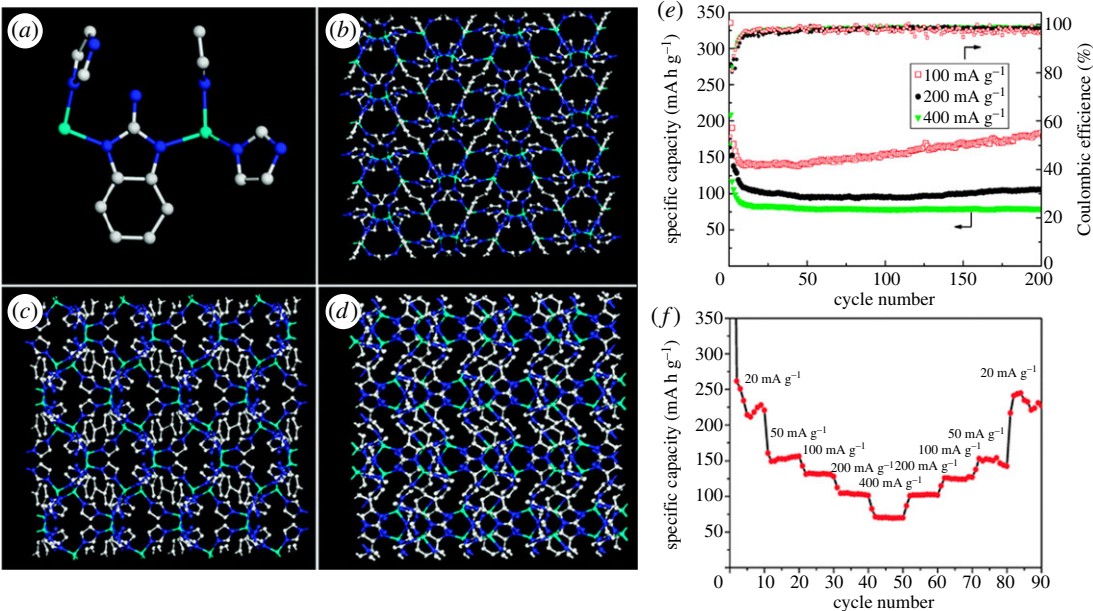

**Figure 7.** The structure of the BMOF: (a) the coordination environment of the Zn²⁺ ion and ligands. (b–d) The structure of the BMOF viewed along the a-axis (b), the b-axis (c) and the c-axis (d), (e) cyclic performance and (f) rate performance of BMOF. Colour scheme: Zn, light blue; N, blue; and C, white [31]. Images reproduced with permission.

charge–discharge cycles at $400 \, \text{mA g}^{-1}$, demonstrating rapid kinetics during cycling. The highly symmetrical pattern surface shown in figure 13 facilitates ion diffusion in interconnected three-dimensional channels even at high-speed cycling while maintaining the robustness of the MOF structure.

Kulandainathan *et al.* [32] synthesized a new anode material $Cu_2(C_8H_4O_4)_4$(Cu-BDC), which is a Cu-based MOF. Studies have shown that Cu-BDC has the ability to store reversible lithium ions, which delivers a stable capacity of $227 \, \text{mA h g}^{-1}$ in the first cycle, which is about 95% of the theoretical capacity. Han *et al.* [33] synthesized a new type of MOF, metal-organic nanofibres (MONF), using amino acids and copper nitrate as precursors. General cyclic capacity under the voltage window of 0.01–3 V and excellent cycle stability are obtained. The reversible capacity of MONF after about 200 cycles is approximately $233 \, \text{mA h g}^{-1}$ at $50 \, \text{mA g}^{-1}$. Zheng *et al.* [34] reported that Ni–Me₄bpz can be synthesized using 3,3′5,5′-tetramethyl-4,4′-bipyrazole(H₂Me₄bpz) and nickel

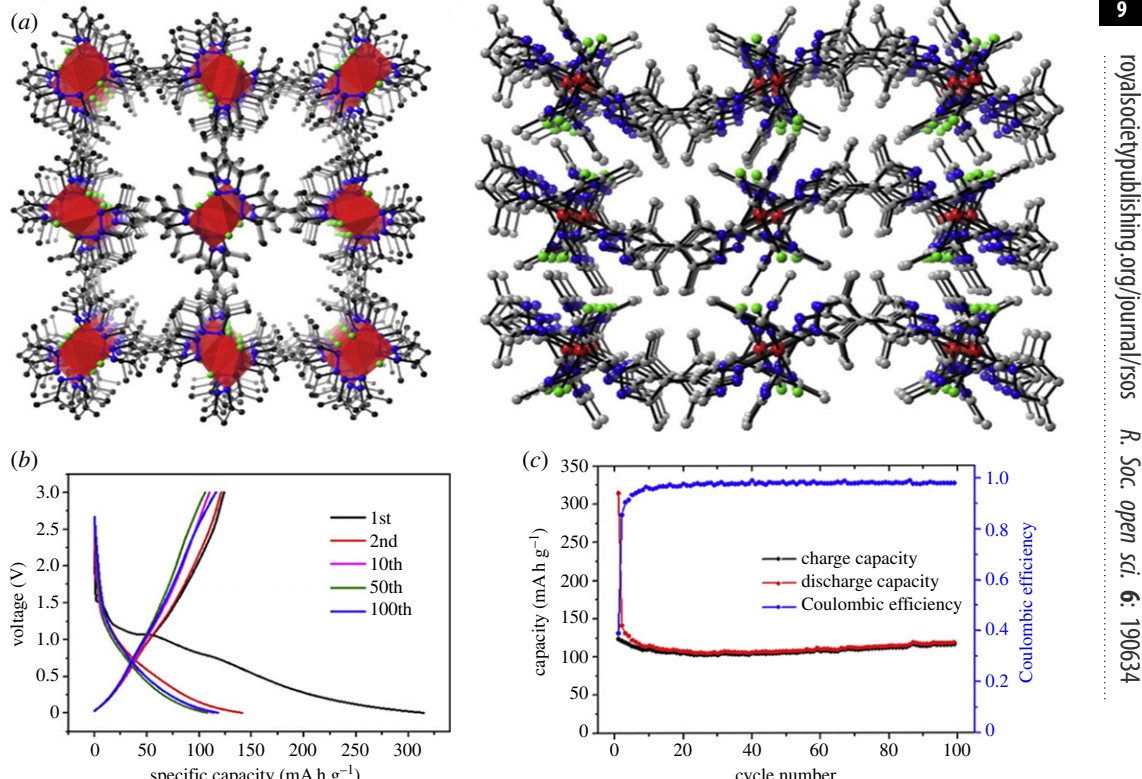

**Figure 8.** (a) The structure of the Ni–Me$_4$bpz, (b) charge–discharge curve of Ni–Me$_4$bpz at a current density of 50 mA g$^{-1}$ and (c) cycle performance of Ni–Me$_4$bpz [34]. Images reproduced with permission.

chloride hexahydrate by the solvothermal method. As shown in figure 8, the initial discharge capacity was 320 mA h g$^{-1}$ at 50 mA g$^{-1}$, the second discharge capacity was reduced to 140 mA h g$^{-1}$, and the capacity loss was about 56.25%. This capacity loss is mainly due to the irreversible lithium loss caused by the incomplete conversion reaction and SEI layer. The specific capacity of the electrode material of the lithium–ion battery remains stable from the third cycle. After 100 cycles, it is about 120 mA h g$^{-1}$, and the Coulomb efficiency can be maintained above 98%. This shows that it has good cycle stability. The crystal structure test of Ni–Me$_4$bpz indicates that the two-dimensional layered structure and flexible ligand can promote the intercalation/deintercalation of lithium ions and improve the electrochemical stability of Ni–Me$_4$bpz. Mahanty *et al.* [35] reported that a MOF with high capacity and rate performance had been synthesized, and studies have shown its ability to reversibly store lithium. In the voltage range of 0.01–2 V, the high charge–discharge-specific capacity under 0.1 and 1 A g$^{-1}$ is 694 and 400 mA h g$^{-1}$, respectively. It was demonstrated that the conventional conversion reaction of Li storage may not be applicable to Mn-BTC. The conjugated carboxylate is a weak electron-withdrawing body with a stronger $\pi$–$\pi$ interaction, which may replace the lithium storage mechanism. It involves the organic part. The Mn-BTC test shows that MOFs can have a very high lithium storage capacity by appropriately selecting an organic ligand, and the redox reaction may not involve a conversion mechanism. On the contrary, the organic part of the MOF seems to play a vital role.

# 3. MOF-derived electrode materials for lithium–ion batteries

## 3.1. MOF-derived metal oxides

Conductivity and stability are the main obstacles in the application of MOFs in battery systems. MOF-derived electrode materials and MOF-based composite materials can effectively solve these problems. Here, we mainly introduce MOF-derived electrode materials.

There are many kinds of electrode materials derived from MOFs for lithium–ion batteries, including N-containing derivative electrode materials of MOFs, S-containing derivative electrode materials of MOFs and P-containing derivative electrode materials of MOFs, among which the most representative

**Table 2.** Electrode materials of lithium–ion batteries prepared by using MOFs as precursors. RC, reversible capacity (mA h g$^{-1}$); CN, cycle number; TC, theoretical capacity.

| MOFs | sample | rate (C or mA g$^{-1}$) | TC (mA h g$^{-1}$) | RC (mA h g$^{-1}$) | CN | Refs. |
|---|---|---|---|---|---|---|
| MIL-88-Fe | $\alpha$-Fe$_2$O$_3$ | C/0.2 | 1000 | 911 | 50 | [53] |
| Fe-MOF | $\alpha$-Fe$_2$O$_3$ | 100 | 1000 | 1024 | 40 | [54] |
| Mn$_3$[Fe(CN)$_6$]$_2$·$n$H$_2$O | Mn$_{1.8}$Fe$_{1.2}$O$_4$ | 200 | 1018 | 827 | 60 | [55] |
| MIL-88-Fe/3DGN | Fe$_2$O$_3$/3DGN | 200 | 1000 | 864 | 50 | [56] |
| MOF-5 | ZnO/ZnFe$_2$O$_4$/C | 500 | 978 | 1390 | 100 | [57] |
| FeC$_2$O$_4$·2H$_2$O | C-Fe$_3$O$_4$ | 100 | 1000 | 975 | 50 | [58] |
| Fe-ZIF | Fe$_2$O$_3$@N-C | C/0.1 | 1000 | 1573 | 50 | [59] |
| Co$_3$(NDC)$_3$(DMF)$_4$ | Co$_3$O$_4$ | 50 | 890 | 965 | 50 | [60] |
| ZIF-67 | Co$_3$O$_4$ | 100 | 890 | 780 | 100 | [61] |
| [Co(HO-BDC)(bbb)] | Co$_3$O$_4$ | 500 | 890 | 852 | 100 | [62] |
| MOF-71 | Co$_3$O$_4$ | 200 | 890 | 913 | 60 | [63] |
| Co-MOF | Co$_3$O$_4$ | 100 | 890 | 470.3 | 90 | [64] |
| ZIF-67/GO | Co$_3$O$_4$ | 100 | 890 | 1550 | 60 | [65] |
| PBA Zn$_3$[Co(CN)$_6$]$_2$ | ZnO/Co$_3$O$_4$ | 100 | 978 | 957 | 100 | [66] |
| Zn-Co-MOF | ZnO/ZnCo$_2$O$_4$ | 2000 | 978 | 1016 | 250 | [67] |
| Ni-MIL-77 | NiO | 100 | 718 | 700 | 60 | [68] |
| Ni$_3$(BTC)$_2$·12H$_2$O | NiO | 100 | 718 | 1019 | 100 | [69] |
| Cu-Ni-BTCMOF | CuO@NiO | 100 | 718 | 1061 | 200 | [70] |
| Co$_3$[Fe(CN)$_6$]$_2$@Ni$_3$[Co(CN)$_6$]$_2$ | Fe$_2$O$_3$@NiCo$_2$O$_4$ | 100 | 1000 | 1079.6 | 100 | [71] |
| Fe$_2$Ni MIL-88@TiO$_2$ | NiFe$_2$O$_4$@TiO$_2$ | 100 | 900 | 1034 | 100 | [72] |
| ZIF-6 | 7 NiCo$_2$O$_4$/NiO | 200 | 1000 | 1535 | 100 | [73] |
| Fe$_2$Ni MIL-88/Fe MIL-88 | NiFe$_2$O$_4$/Fe$_2$O$_3$ | 100 | 1000 | 936.9 | 100 | [74] |
| Ni$_2$[Fe(CN)$_6$] | NiFe$_2$O$_4$ | C/1 | 900 | 975 | 200 | [75] |
| Cu-MOF | CuO | 100 | 670 | 470 | 100 | [76] |
| MOF-199 | CuO | 100 | 670 | 484.2 | 40 | [77] |
| [Cu$_3$(btc)$_2$]n | CuO/Cu$_2$O | 100 | 670 | 740 | 250 | [78] |
| MIL-125 | TiO$_2$ | 1 C | 167 | 166.2 | 500 | [79] |
| Cu-MOF | CuO-G | 60 | 670 | 700 | 40 | [80] |
| MIL-101 | Cr$_2$O$_3$@TiO$_2$ | 0.5C | 670 | 510 | 500 | [81] |

derivative material of MOFs is the metal oxide derived from MOFs. Metal oxides formed by sintering MOFs as sacrificial precursors have been attracting attention as electrode materials. Metal oxides have excellent theoretical capacity and safety, while MOFs are the precursor to regulate the pore size and structure of metal oxides. Therefore, metal oxides prepared by sintering MOFs as precursors are one of the best choices for electrode materials [51,52]. As shown in table 2, we summarize the application of MOF-derived metal oxides as electrode materials for lithium–ion batteries. Tarascon *et al.* [82] found that the equilibrium conversion reaction between nano-scale transition metal oxides and lithium occurs. Its working principle is as follows: $MO + 2Li \leftrightarrow M + Li_2O$ (M = Fe, Co, Ni, Mn, etc.) and provides a higher reversible capacity than the graphite anode. Based on the above reaction principles and concepts, it is possible to use metal oxides as electrode materials. Cho *et al.* [53] reported a spindle-shaped porous $\alpha$-Fe$_2$O$_3$ prepared by calcinating an iron-based MOF. Under special conditions, as shown in figure 9, the reversible specific capacity of porous $\alpha$-Fe$_2$O$_3$ can remain 911 mA h g$^{-1}$ after 50 cycles. Under certain conditions, its stable capacity can even reach 424 mA h g$^{-1}$ at 10 C. Different from the previous $\alpha$-Fe$_2$O$_3$ produced by sintering MIL-88-Fe as a precursor, the Fe-MOF synthesized

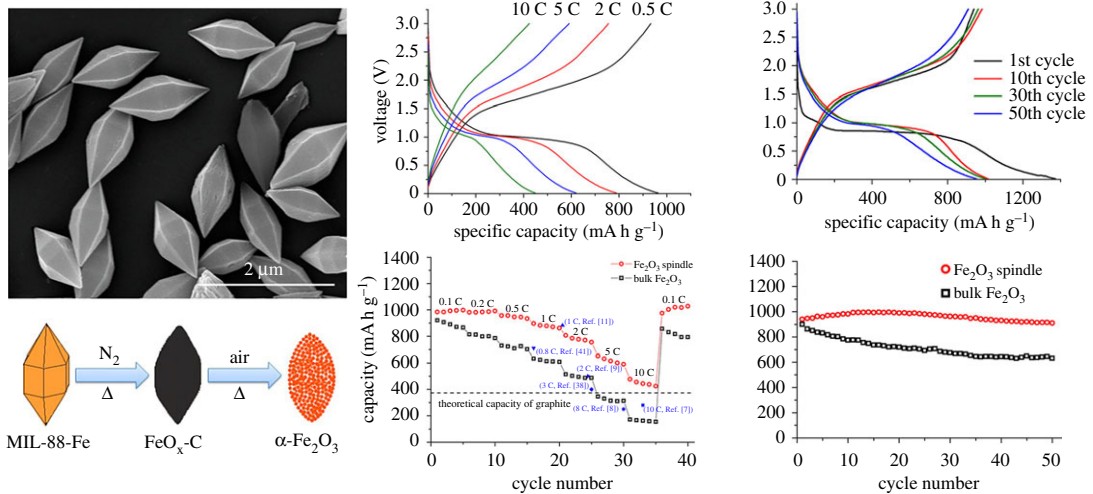

**Figure 9.** SEM image of as-prepared MIL-88-Fe, schematic diagram and electrochemical properties of synthetic spindle-like porous α-Fe$_2$O$_3$ [53]. Images reproduced with permission.

by the solvothermal synthesis of FeCl$_3$ and terephthalic acid is a precursor. The α-Fe$_2$O$_3$ produced by one-step sintering reported by Ogale *et al.* [54] also has good electrochemical properties. The α-Fe$_2$O$_3$ produced by pyrolysis of Fe-MOF as a precursor is an electrode material, and the assembled single cell has a reversible capacity of 1024 mA h g$^{-1}$ under 100 mA g$^{-1}$. In addition, after 40 constant current cycles, the capacity retention rate is over 90%, and the battery characteristics are good under high current circulation, indicating that the battery has excellent cycle performance. Chen *et al.* [55] reported a porous nanocube Mn$_{1.8}$Fe$_{1.2}$O$_4$ containing Mn and Fe. Like other metal oxides produced by MOF as a precursor, Mn$_{1.8}$Fe$_{1.2}$O$_4$ inherits the morphology of its precursor, and the specific surface area is relatively high because of the release of carbon dioxide and nitrogen dioxide. The prepared porous Mn$_{1.8}$Fe$_{1.2}$O$_4$ shows greatly improved lithium storage performance. With porous Mn$_{1.8}$Fe$_{1.2}$O$_4$ as the anode material, it has a capacity of 827 mA h g$^{-1}$ after 60 cycles at 200 mA g$^{-1}$, and has high capacity and cycle stability. The improvement of the performance of the anode is attributed to the porous structure of MOFs and the large number of mesopores of MOFs. These mesopores effectively improve the stability of the electrode material structure, reduce the diffusion of lithium ions and reduce the expansion of buffer volume of lithium ions in the lithium ions' insertion and deintercalation process, indicating that Mn$_{1.8}$Fe$_{1.2}$O$_4$ has excellent electrochemical properties.

Zhang *et al.* [56] developed a new method for preparing composite materials coated with metal oxides and graphene, with MOFs as a precursor. This MOF is first synthesized on the three-dimensional graphene network as a precursor of the metal oxide, and then the metal oxide/3DGN composites were obtained by the two-step heat treatment. As shown in figure 10, the capacity of Fe$_2$O$_3$ decreases to 261 mA h g$^{-1}$ through 50 charge–discharge cycles at 0.2 A g$^{-1}$. The stable reversible capacity of Fe$_2$O$_3$/3DGN composites is 864 mA h g$^{-1}$, indicating that the Fe$_2$O$_3$/3DGN composite has good recyclability. Compared with the Fe$_2$O$_3$/3DGN electrode, the Fe$_2$O$_3$ electrode exhibits the same poor performance in rate performance. Huang *et al.* [57] synthesized hybrid hollow porous octahedral Fe$^{III}$-MOF-5 based on the method given in the literature, the improved reflux method. As shown in figure 11, the final product of the hollow porous ZnO/ZnFe$_2$O$_4$/C octahedron is then obtained by heat treatment. Experiments show that the electrode material has a capacity of 762 mA h g$^{-1}$ at 10 A g$^{-1}$, which is more than twice the capacity of graphite. Lu *et al.* [58] prepared a porous carbon-modified iron oxide by pyrolysis using ferrous oxalate anhydrate as a precursor. Fe$_3$O$_4$ nanoparticles were prepared by the precursor method, the porous structure of the precursor provides a large void space, which is conducive in reducing the volume change during the lithium ions' insertion/extraction process. The residual carbon network not only helps to form a stable SEI layer on the surface of the electrode but also helps to enhance conductivity. When the current density of C-Fe$_3$O$_4$ is 100 mA g$^{-1}$, through 50 cycles, it still has a high reversible capacity of 975 mA h g$^{-1}$, which indicates that it has an excellent capacity and cycle stability.

Chen *et al.* [59] prepared the Fe-based zeolitic imidazolate skeleton by the coprecipitation method. Then, the Fe-based zeolitic imidazolate skeleton is used as a precursor to obtain pure Fe$_2$O$_3$ by the two-step dehydration treatment. This material is electrochemically tested as an anode material. Its

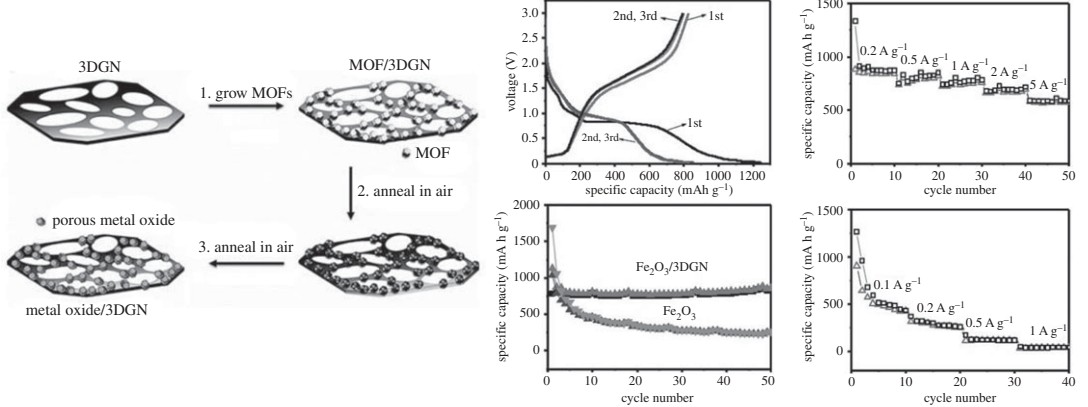

**Figure 10.** Schematic illustration of the process used for the synthesis and electrochemical properties of metal oxide/3DGNcomposites [56]. Images reproduced with permission.

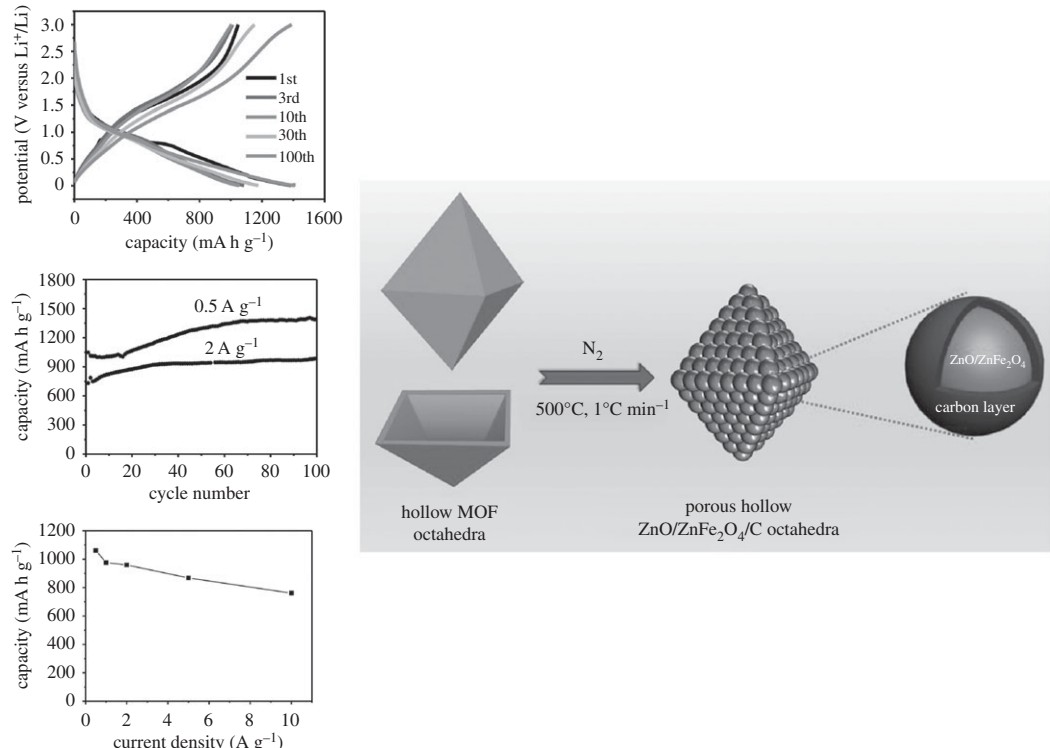

**Figure 11.** The preparation process and electrochemical properties of porous hollow ZnO/ZnFe$_2$O$_4$/C octahedral [57]. Images reproduced with permission.

charge and discharge capacities can reach 1573 mA h g$^{-1}$ after 50 cycles at 1000 mA g$^{-1}$. Its charge and discharge capacities can be stabilized at 1142 mA h g$^{-1}$ under 100 cycles. Similar to Fe$_2$O$_3$, Co$_3$O$_4$ is the most promising electrode material and can be pyrolysed from many Co-based MOFs as precursors. Xu *et al.* [60] obtained Co$_3$O$_4$ by pyrolysis a porous precursor (Co$_3$(NDC)$_3$(DMF)$_4$) with good morphology. The Co$_3$O$_4$ prepared by this method has a small size of only 25 nm and is easily agglomerated to form an agglomerated secondary structure having a diameter of about 250 nm. This unique structure is helpful to improve the electrode capacity and prolong the cycle life. The first discharge capacity of the Co$_3$O$_4$ reaches 1118 mA h g$^{-1}$ when the current density is 50 mA g$^{-1}$, and the first cycle retention rate reaches 75% in the first cycle. Then the cycle capacity gradually increased to 965 mA h g$^{-1}$ after the fifth cycle, which is 86% of the initial capacity. This characteristic is due to the unique structural characteristics of Co$_3$O$_4$ aggregates. Huang *et al.* [61] prepared the cobalt-containing ZIF-67 template with rhombic

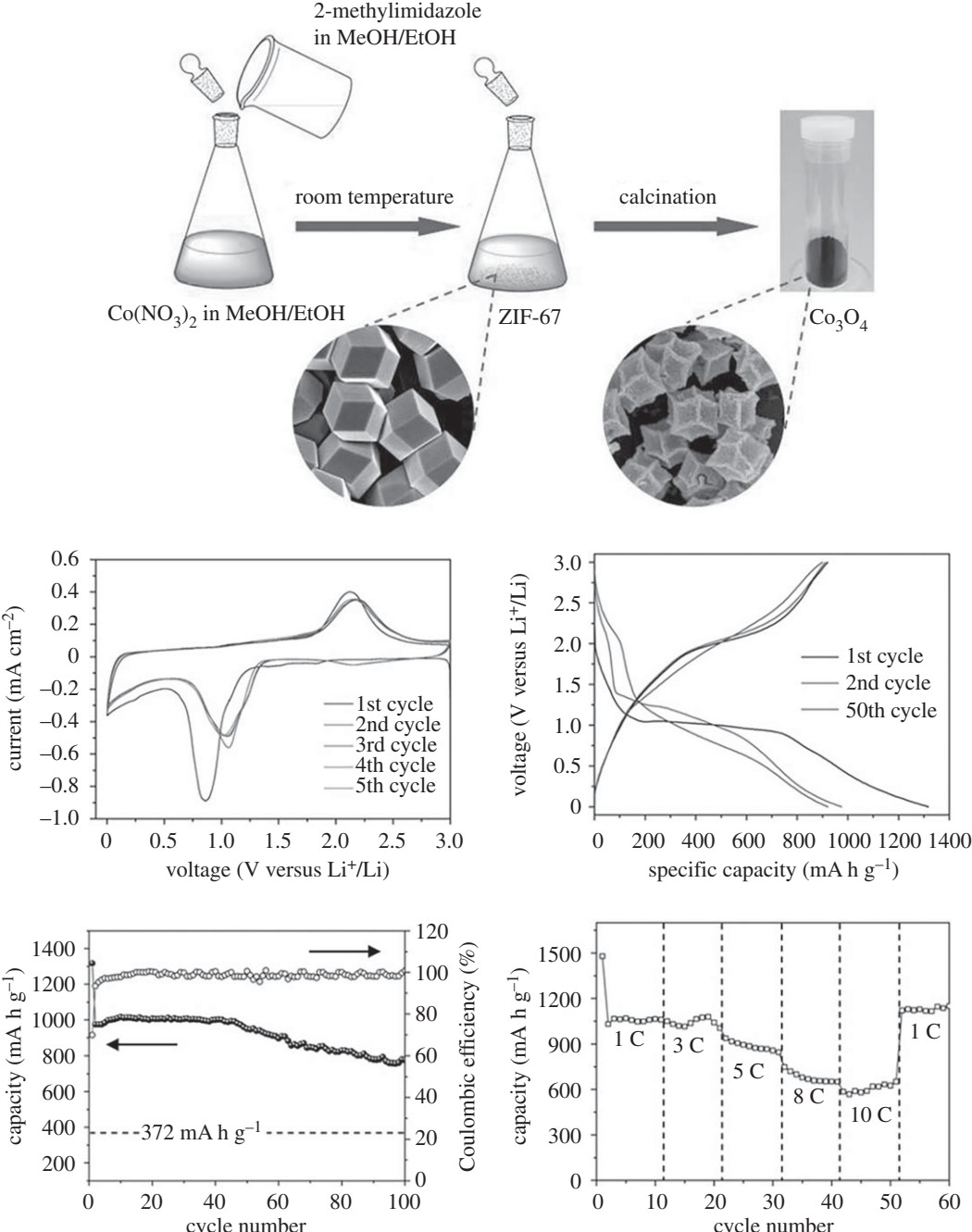

**Figure 12.** Schematic illustration of the fabrication and electrochemical properties of porous $Co_3O_4$ hollow dodecahedra [61]. Images reproduced with permission.

dodecahedron morphology, and then the two-step heat treatment was carried out at a medium temperature to obtain $Co_3O_4$ with the precursor ZIF-67 structure. As shown in figure 12, $Co_3O_4$ hollow dodecahedron has first capacities of 1317 and 921 mA h g$^{-1}$ at 100 mA g$^{-1}$, respectively. For the first time, Wen *et al.* [62] proved the application of new layered $Co_3O_4$ nanoplates synthesized by chemical conversion using layered cobalt-based CP as the template in lithium–ion battery electrode materials. Coordination of two-dimensional layered texture features $Co_3O_4$ nanosheets has significantly enhanced lithium storage capacity, including high-capacity lithium storage at 852 mA h g$^{-1}$ under 500 mA g$^{-1}$, good rate performance of 597 mA h g$^{-1}$ at 1000 mA g$^{-1}$ after 100 cycles, and up to 200 cycles. MOF-71 is another precursor for the preparation of mesoporous nanostructured $Co_3O_4$, MOF-71 was synthesized by Hu *et al.* [63] by a solvent method and then pyrolysed to obtain $Co_3O_4$ having a porous structure. The reversible chemical reaction of $Co_3O_4$ with lithium ion with the precursor MOF-71 structure can be given as follows:

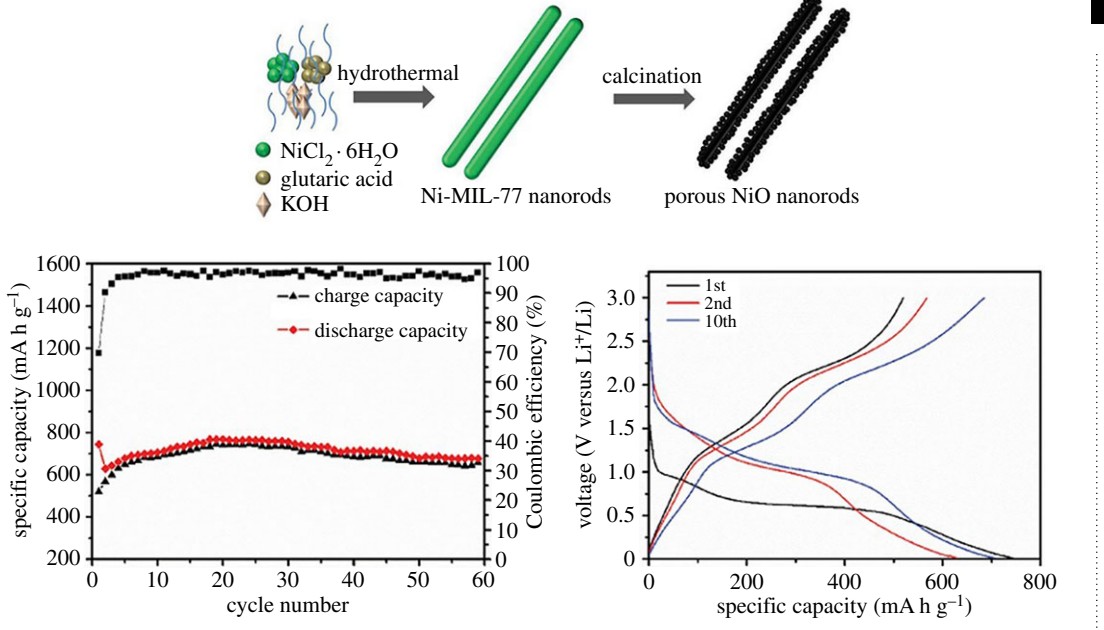

**Figure 13.** Schematic diagram of the process used to synthesize the porous NiO nanorods and the electrochemical performance of the porous NiO nanorods [68]. Images reproduced with permission.

(1) $Co_3O_4 + 8Li^+ + 8e^- \rightarrow 3Co + 4Li_2O$ (first lithiation)
(2) $Co + Li_2O \rightarrow CoO + 2Li^+ + 2e^-$ (subsequent delithiation/lithiation)

The first cyclic reversible capacities of MOF-71@300N are 1286.1 and 879.5 mA h g$^{-1}$ under 200 mA g$^{-1}$. This study proposes a new idea for the preparation of mesoporous nanostructures of metal oxide electrodes.

Yang *et al.* [64] reported that a new Co-based metal-organic framework (Co-MOFs) is pyrolysed into porous $Co_3O_4$, which is better than the flower-like $Co_3O_4$ for lithium–ion batteries, which may be due to smaller particles. With dimensions and a dense pore structure with a suitable pore size, the first charge capacity of this $Co_3O_4$ is about 1003.5 mA h g$^{-1}$ and Coulomb efficiency is about 75.7%. The irreversible capacity in the first cycle is generally due to the decomposition of electrolytes into SEI layers and interfacial lithium storage, which is common in most electrode materials as mentioned earlier. Wang and his team's research [68] proposes a simple, scalable way for the synthesis of porous NiO nanorods by heat treatment of MOFs in air. As shown in figure 13, when the synthesized NiO is used as an anode material, the first discharge has 743 mA h g$^{-1}$ under 100 mA g$^{-1}$, and the high stable capacity remains about 700 mA h g$^{-1}$ through 60 reversible cycles. The one-dimensional mesoporous NiO nanorods reported by Wang *et al.* [69] were first synthesized from a rod-like nickel-molybdenum precursor at room temperature. Mesoporous NiO nanorods provide theoretical reversible capacity greater than 1019 mA h g$^{-1}$ after 100 cycles at 100 mA g$^{-1}$, which shows excellent performance as electrode materials. Mesoporous structures derived from MOF precursors and nanorod morphology integrated with one-dimensional nanoparticles promote lithium diffusion, resisting volume change and maintaining the stability of the electrode structure during repeated cycles resulting in reversible lithium storage capacity and good cycling capacity.

Also, by annealing, CuO prepared with Cu-based MOF as a template is also one of the most promising for anode materials. Huang *et al.* [76] synthesized porous CuO hollow octahedron by a simple method. The experimental steps included hydrothermal preparation of the copper BTC MOF template and thermal decomposition of the precursor at 300°C. The first discharge capacity of the CuO is 1208 mA h g$^{-1}$ when the current density is 100 mA g$^{-1}$. The initial Coulomb efficiency is 40%, the reason for this relatively low level could be interfacial lithium storage, organic conductive polymers formed by irreversibly inserting/leaving lithium ions into the main structure to form SEI layers and adsorption of impurities on CuO. It is worth noting that the CuO hollow octahedron prepared by several discharge–charge cycles exhibited excellent ring capacity retention and Coulomb efficiency stability still higher than 99%. The reversible capacity of 470 mA h g$^{-1}$ can be stable up to

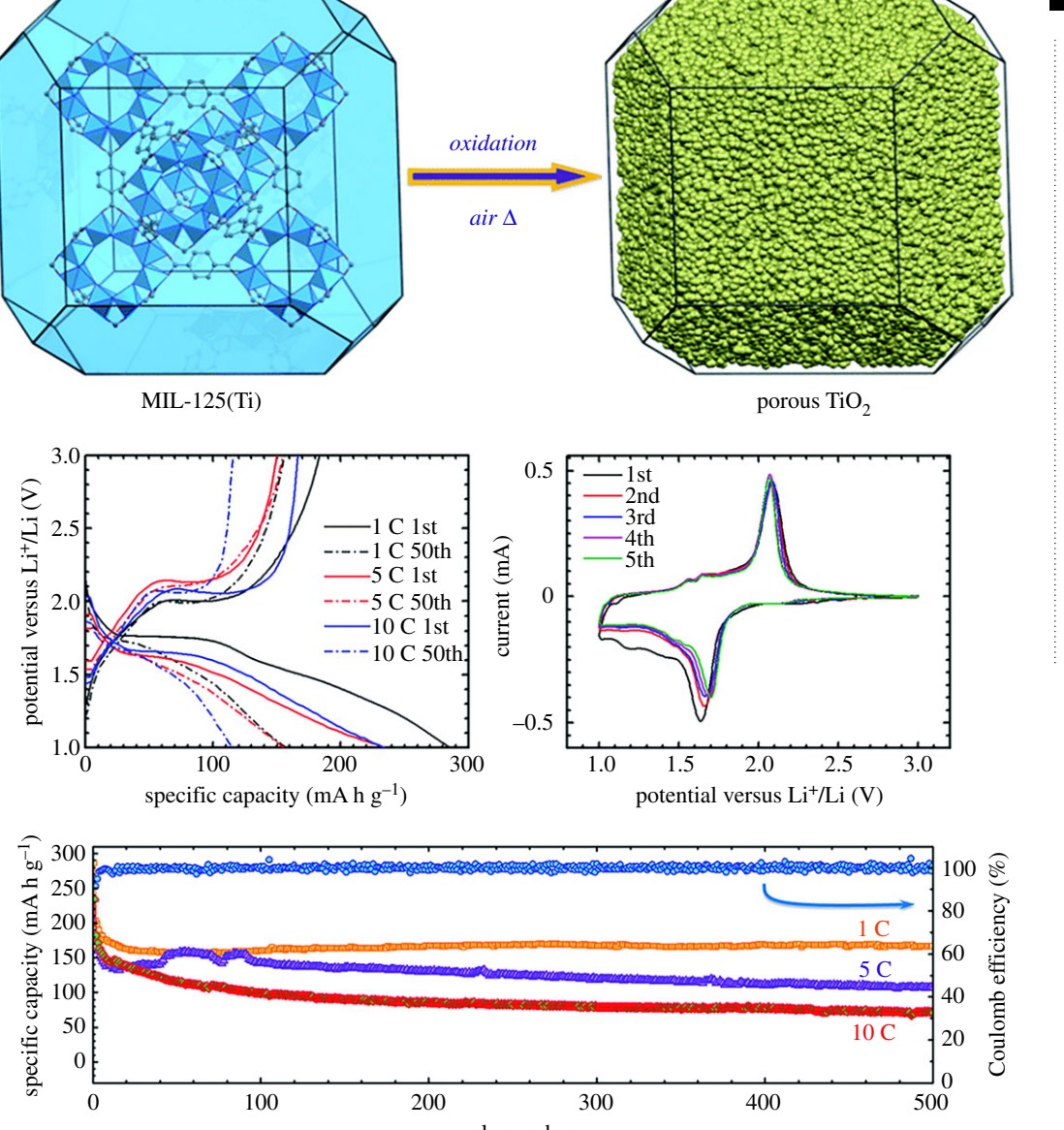

**Figure 14.** Schematic illustration for the synthesis and the electrochemical performance of porous TiO$_2$ from a MOF precursor [79]. Images reproduced with permission.

100 cycles in electrochemical testing. The improved cyclicity is mainly due to the unique characteristics of the porous CuO hollow octahedron. The above mechanism indicates that the synthesized CuO octahedron exhibits good performance, has very stable cycle behaviour and good rate performance.

Ogale *et al.* [77] reported the synthesis of CuO nanostructures based on MOF's method. Spherical and phase-pure CuO nanoparticles were prepared by pyrolysis of copper-based MOF precursors. Ogale and his team used MOF-199 as a precursor to obtain CuO with a precursor porous structure by pyrolysis. The electrochemical cycling performance of the compound as an electrode material was tested. Results found that the reversible capacity of 538 mA h g$^{-1}$ (1.6 mol Li) can be obtained. After 40 constant current cycles, the Li/CuO semi-battery maintained about 90% first reversible capacity and could provide 210 mA h g$^{-1}$ reversible capacity at 2 A g$^{-1}$. Qian *et al.* [79] first prepared porous anatase TiO$_2$ by calcining a MOF MIL-125(Ti) precursor in an air atmosphere at 380°C. As shown in figure 14, the porous anatase titanium dioxide is prepared with MIL-125(Ti) as the precursor of the anode has good capacity retention and rate performance. The porous structure of anatase titanium dioxide enhances the diffusion of lithium ion and reduces the influence of the volume change of the electrode. The electrochemical tests showed that the stable capacities of the electrode were 166, 106 and 71 mA h g$^{-1}$ at 1, 5 and 10 C under 500 cycles.

In addition to these metal oxides prepared from MOFs as precursors for lithium–ion batteries, there are also metal oxide composites prepared by combining a plurality of MOFs as precursors or MOFs with other materials. For example, $Co_3O_4$ [65], $ZnO/Co_3O_4$ [66], $ZnO/ZnCo_2O_4$ [67], $CuO@NiO$ [70], $Fe_2O_3@NiCo_2O_4$ [71], $NiFe_2O_4@TiO_2$ [72], $NiCo_2O_4/NiO$ [73], $NiFe_2O_4/Fe_2O_3$ [74], $NiFe_2O_4$ [75], $CuO/Cu_2O$ [78], CuO-G [80] and $Cr_2O_3@TiO_2$ [81]. By using metal oxide composite materials prepared with MOFs as precursors for electrode materials, the electrochemical stability and capacity have been greatly improved.

## 3.2. MOF-derived other electrode materials

In addition to metal oxides derived from MOFs, other N-, S- or P-containing derivatives of MOFs can be collectively referred to as porous carbon materials derived from MOFs, which are widely used in electrode materials due to their excellent electrochemical capacity and good cyclic stability.

Xu *et al.* [81] reported for the first time that derivatives of MOFs were used in electrochemistry. Porous carbon with precursor morphology was synthesized by pyrolysis of the precursor FA in pore MOF-5 by the precursor method. The obtained porous carbon derivatives can be used as electrodes and have excellent electrochemical properties. Wang *et al.* [83] obtained $C_OS_X$ composites coated with rGO by thermal vulcanization of MOF/GO precursors and used them as anode materials. The electrochemical tests show that the electrode material has excellent electrochemical properties. When the current density was 100 mA g$^{-1}$, the first specific capacities were 1248 and 1320 mA h g$^{-1}$, respectively. After 100 charge–discharge cycles, the specific capacities were still 670 and 613 mA h g$^{-1}$. $C_OS_X$/rGO composites have good conductivity and a suitable porous structure, which can promote the migration of lithium ions and reduce the internal stress during charging and discharging, thus significantly improving the electrochemical properties of electrode materials.

Chen *et al.* [84] synthesized nitrogen-doped graphene electrode materials with outstanding lithium–ion storage properties. The nitrogen content of most of the nitrogen-doped carbon materials used is about 10 wt%. Nitrogen-doped graphene prepared by pyrolysis of the nitrogen-containing imidazole molecular sieve framework at 800°C in nitrogen atmosphere has a nitrogen content of 17.72 wt%. Graphene particles show excellent electrochemical properties after nitrogen doping. The capacity of the electrode is 2132 mA h g$^{-1}$ with 100 mA g$^{-1}$ after 50 cycles and 785 mA h g$^{-1}$ after 1000 cycles under 100 mA g$^{-1}$.

Porous carbon derived from MOFs is porous carbon material formed by carbonization of MOFs. With advantages such as special morphology, large surface area, unique pore size distribution and remarkable performance, MOF-derived materials have been extensively used in sustainable energy and alternative clean technologies. MOFs and MOF-derived materials are one of the best candidates for next-generation lithium–ion batteries.

# 4. MOFs as electrode materials for lithium–sulfur batteries

As one of the most outstanding candidates for energy storage systems, lithium–sulfur batteries have attracted much attention recently. Compared with lithium–ion batteries, lithium–sulfur batteries not only have higher theoretical capacity (1675 mA h g$^{-1}$) but also because of being an abundant natural S resource, low cost and environmental protection [85–87]. As shown in figure 15, the principle of electrochemical conversion in Li–sulfur batteries, that is, the operation of S reduction is as follows [88]:

$$S_8 + 2e^- \rightarrow S_8^{2-}$$
$$S_8^{2-} \leftrightarrow S_6^+ + \frac{1}{4}S_8$$
$$2S_6^{2-} + 2e^- \leftrightarrow 3S_4^{2-}$$
$$3S_4^{2-} + 2e^- \leftrightarrow 4S_3^{2-}$$
$$S_3^{2-} + 6Li^+ + 2e^- \leftrightarrow 3Li_2S_2$$
$$Li_2S_2 + 2Li^+ + 2e^- \leftrightarrow 2Li_2S$$

Although lithium–sulfur batteries have these advantages, there are many factors that limit their commercial development. First, the electrical insulation of S and $Li_2S$ and $Li_2S_2$ makes the utilization of active materials low. Secondly, 80% volume change is accompanied by the electrochemical reaction. Also, the polysulfide intermediate $Li_2Sn$ ($3 \leq n < 8$) easily diffuses into the organic liquid electrolyte, resulting in rapid dissolution of lithium–sulfur battery capacity and low Coulombic efficiency [89]. Of course, many methods have been proposed to solve these problems, such as sulfur and sulfide are

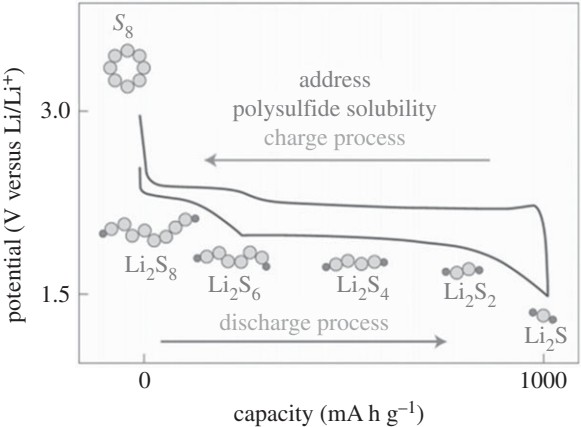

**Figure 15.** Schematic and voltage profiles of a Li-S cell [5]. Images reproduced with permission.

encapsulated or embedded in porous carbon materials by physical adsorption, strengthening sulfur species by chemical adsorption with metal oxide additives and physically blocking sulfur with conductive polymers [90–94]. But these effects are not very obvious, so we hope to use MOFs to improve the energy conversion rate and reduce the capacity loss rate of lithium–sulfur batteries.

## 4.1. MOF-based electrode materials for lithium–sulfur batteries

The dissolution and fixation of sulphur and polysulfides by MOFs with large pore volume, adjustable structure and super high specific surface area, as well as how to design and store energy reasonably, have broad application and developmental prospects.

The application of MOFs in lithium–sulfur batteries directly includes three steps: first, mixing MOFs with sulfur, then heat treatment and the third step is to use the active materials after heat treatment as the electrode material of lithium–sulfur batteries. When sulfur is embedded in the macropore of MOFs, the unique pore of MOFs plays an important role in the packaging and conversion of sulfur and sulfide.

Deng et al. [95] reported a novel method to increase the electrochemical conductivity of MOFs by five to seven times. Therefore, this method can effectively improve the electrochemical capacity and cycle stability when MOFs are used as active substances for electrode materials. As shown in figure 16, their method combines the advantages of MOFs and the conductivity of conductive polymers. Their main process of synthesizing materials included first the establishment of a ppy MOF chamber by polypyrrole (ppy) to limit sulfur in lithium–sulfur batteries, then through comparing ppy film boxes with different pore structures, the key role of ion diffusion in high-speed performance was proved for the first time. ppy-S-in-PCN-224 electrodes with MOF's characteristics still have high reversible specific capacities of 670 and 440 mA h g$^{-1}$ through 200 and 1000 cycles at 10 C. The results show that the size and aperture of MOFs have a great influence on the charge and discharge.

Whether it is a lithium–ion battery or a lithium–sulfur battery, the particle size of the active materials in the battery electrode materials has an important influence on battery performance. For lithium–sulfur batteries, porous MOFs are used to store sulfur and inhibit the transformation of polysulfides; however, the particle size of MOFs used in lithium–sulfur batteries as an important parameter has not been explored to a large extent. Li et al. [96] reported five groups of ZIF-8 samples with different particle sizes (from less than 20 nm to less than 1 mm) were synthesized and used as the S@MOF cathode. The electrochemical performance test showed that ZIF-8 had the potential for sulfur storage. The results show that increasing the utilization of sulfur in lithium–sulfur batteries can be achieved by reducing the particle size of ZIF-8. The ZIF-8 particle size reaches an electrochemical capacity of more than 950 mA h g$^{-1}$ at 0.5 C at less than 20 nm, and achieves optimum cycle stability of 0.5 C over 200 cycles with a capacity of 75% at a medium size of around 200 nm.

Tarascon et al. [97] proposed a measure based upon the shortcomings of soluble polysulfide formation. This strategy is using MIL-100(Cr) as the sulfur-impregnated main part. As shown in figure 17, it was found that the electrode containing sulfur impregnated in the pores of the MOF showed a significant reducing capacity loss rate of the sulfur cathode. Experimental testing demonstrates the recycle of polysulfides. MIL-100(Cr)/S@RT + 50%C discharge capacity increases to 1580 mA h g$^{-1}$ and its capacity retention rate has also improved considerably.

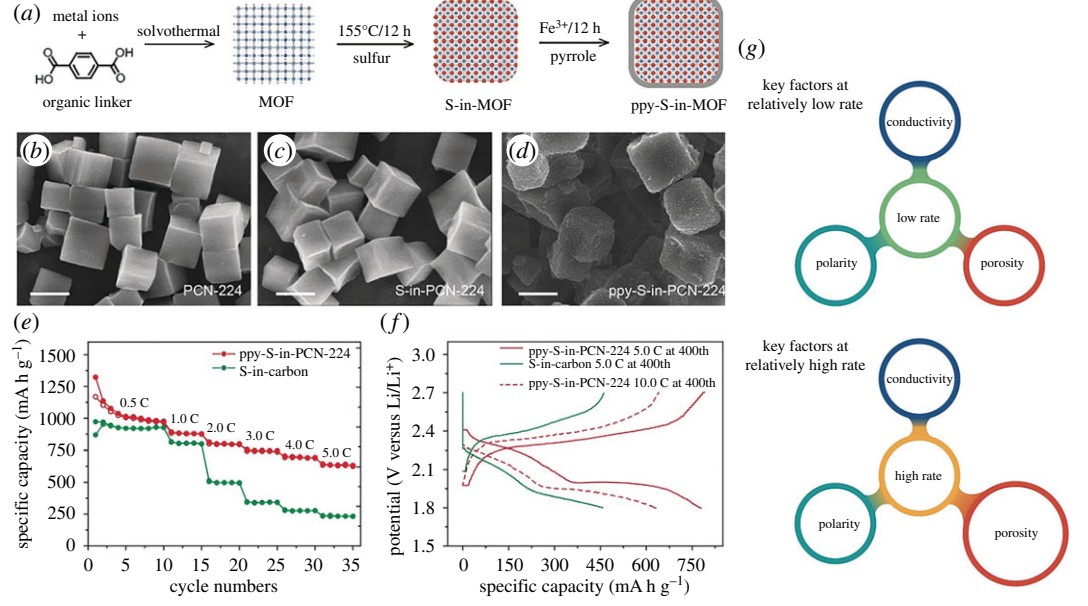

**Figure 16.** (*a*) Synthetic procedure of ppy-S-in-MOF constructs, SEM of (*b*) PCN-224, (*c*) S-in-PCN-224 and (*d*) ppy-S-in-PCN-224, (*e*) and (*f*) electrochemical performance of ppy-S-in-PCN-224, (*g*) key factors at a different rate [95]. Images reproduced with permission.

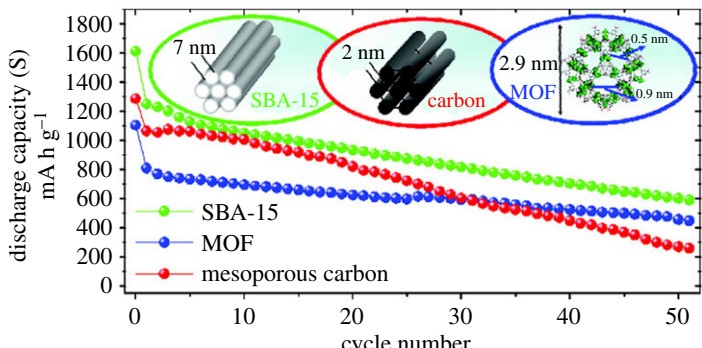

**Figure 17.** Cycling performance of the composites in 1 M of LiTFSI in the TMS electrolyte with C/10 current density for the voltage range between 1.0 and 3.0 V versus Li [97]. Images reproduced with permission.

Xiao *et al.* [98] reported that the novel nickel-based metal-organic framework (Ni-MOF) can significantly immobilize the physical and chemical interactions of polysulfides in the cathode structure through molecular levels. Through 100 cycles at 0.1 C, the capacity holding rate reached 89%.

## 4.2. MOF-derived carbon electrode materials for lithium–sulfur batteries

In addition to MOFs being directly used as electrode materials, the MOF-derived materials, represented by the electrodes prepared with MOFs as precursors, are also widely used in lithium–sulfur batteries.

Li *et al.* [99] presented the design and synthesis of multi-walled carbon nanotubes@mesoporous carbon (MWCNT@Meso-C) and its use in high-performance lithium–sulfur batteries. Applied in lithium–sulfur batteries by the unique multi-walled carbon nanotubes@metal-organic framework (MWCNT@MOF-5) as a precursor, active sulfur is packaged to the MWCNT@Meso-C matrix prepared by the carbonized MWCNT@MOF-5 polyhedron. Electrochemical tests show that the initial cycle discharge capacity of the MWCNT@Meso-C/S sulfur cathode is 1343 mA h g$^{-1}$ under 0.5 C, and still has a cyclic discharge-specific capacity of 540 mA h g$^{-1}$ through 50 cycles. These manifest that MWCNT@Meso-C is an attractive sulfur cathode material in lithium–sulfur battery applications.

Layered nitrogen-doped MOFs with different pore sizes were synthesized by Li & Yin [100]. A large number of microporous MOFs can effectively fasten a large number of sulfur and sulfur-containing compounds. Nitrogen doping can also improve the sulfur environment, thus enhancing the interaction

between carbon and sulfur. When carbon–sulfur is used as cathode electrode materials of lithium–sulfur batteries, it shows excellent battery performance; after the fifth cycle, the stable capacity of 1008.7 mA h g$^{-1}$ is still displayed. Even at 335 mA g$^{-1}$, the stable capacity of 936.5 mA h g$^{-1}$ is maintained after the 100th charge–discharge cycle, showing a capacity retention rate of 82.6%.

Zhang *et al.* [101] obtained a new potato-like layered porous carbon through the construction of aluminium carbide-MOF; sulfur mainly infiltrated into micropore and mesopore, while macropore was conducive to the transport of lithium ion (Li$^+$). Based on this concept, even if LiNO$_3$ is not used as an additive, lithium–sulfur batteries not only have a high first discharge capacity close to 1200 mA h g$^{-1}$ under 0.1 C, but also have a capacity loss rate of 32% through 200 cycles under 0.5 C, showing good cycle performance and high specific capacity.

Cheetham *et al.* [102] proposed a sulfur-loaded carbonized MOF for the cathode structure of lithium–sulfur batteries. Four zinc-containing MOFs were selected to synthesize unique carbon materials with different pore structures after heat treatment. It is found that the cathode materials derived from MOF with a high mesoporous (2–50 nm) volume have higher initial discharge capacity, while the carbon with a high microporous volume has better cyclic stability. MOFs are also used in all aspects of lithium–sulfur batteries because of the characteristics of MOFs [5,103–107].

In addition to the application of MOFs and MOF-derived materials in lithium–ion batteries and lithium–sulfur batteries, MOF-based composites are also widely used in lithium–ion batteries and lithium–sulfur batteries [108–110]. MOF-based composites are of great help in changing the conductivity and stability of the initial form of MOF. MOF-based composites not only have the structural and functional characteristics of MOFs but also can enhance the performance of materials through the modification and other methods. MOF-based composites play an important role in the modification of MOF-based electrode materials.

# 5. Conclusion

The abundant unique pores of MOFs can supply lithium–ion storage places and enhance the stability of batteries; the central metal ions of MOFs can be treated as active centres of chemical reactions during the charging and discharging of batteries. The interaction between sulphur or sulphide and MOFs makes MOFs the main material of the S cathode for lithium–sulfur batteries due to the structural characteristics of MOFs. The application of MOFs in lithium–ion batteries and lithium–sulfur batteries has proven to be highly promising; the environmental protection and energy saving features can be fully reflected in the application of lithium–ion batteries and lithium–sulfur batteries. The MOF's controllable and flexible structure and porous high specific surface area not only can improve the cycle stability of lithium–ion batteries but also can increase the capacity of lithium–ion batteries, these characteristics of MOFs also inhibit the conversion reaction of polysulfides in lithium–sulfur batteries and increase the battery capacity and cycle stability.

However, challenges and opportunities coexist. MOFs have many problems in the application of lithium–ion batteries and lithium–sulfur batteries. First of all, the stability and the conductivity of MOFs need to be improved in order to be better applied to electrochemical energy storage; many MOFs cannot be stored for a long time in a humid environment, which will affect the cycle stability and charge–discharge-specific capacity of the batteries. Secondly, the composite electrode material can greatly improve the electrochemical performance of the batteries, so the development of composite electrodes containing MOFs needs the exploration of researchers. Finally, the application of MOFs in the field of electrochemistry is not limited to lithium–ion and lithium–sulfur batteries, but also includes supercapacitors and metal–air batteries, fuel cells, which are widely used in the field of electrochemistry and have good development prospects.

Summarizing, we have reported some applications of MOFs in lithium–ion and lithium–sulfur batteries. The development prospects of MOFs are generally optimistic. I believe that through our joint efforts, the industrialization of MOF's applications on lithium–ion and lithium–sulfur batteries will definitely be on the agenda.

Data accessibility. This article has no additional data.

Authors' contributions. J.P.Z., X.H.W. and X.X.Z. formulated the content and structure of the manuscript. J.P.Z. collated and drafted the manuscript. X.H.W. and X.X.Z. provided input for the revision. J.P.Z. assisted in sorting out data obtained from the literature and preparation of maps. All authors finally approved the publication.

Competing interests. The authors declare no competing interests.

Funding. This work was supported by the National Science Foundation of China (grant no. 21373074), startup foundation of Hefei University of Technology (XC2016HGCX001).

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
