## [Reviewer comments · Royal Society Open Science]

Review History

RSOS-190634.R0 (Original submission)

Review form: Reviewer 1

Is the manuscript scientifically sound in its present form?

Yes

Are the interpretations and conclusions justified by the results?

Yes

Is the language acceptable?

Yes

Is it clear how to access all supporting data?

Yes

Do you have any ethical concerns with this paper?

No

Have you any concerns about statistical analyses in this paper?

No

Recommendation?

Accept with minor revision (please list in comments)

Comments to the Author(s)

In this manuscript, the authors summarized the research and application of MOFs in the area of lithium-ion batteries and lithium-sulfur batteries. The content of the manuscript is of logic and significance. But there are some problems as follows:

(1) The authors wrote that "Zhou et al [101] proposed a compartment with MOFs, which plays an important role in ion separation in lithium-sulfur batteries. Its working principle is shown in Figure 31." But I don't think this part belongs to MOF-derived carbon electrode materials for lithium-sulfur batteries. It should be MOF-based separator for lithium-sulfur batteries. The authors can deal with them separately and add some similar content and tables for better comparison.

(2) Many figures are chosen with no logic. The structure and electrochemical performance should be shown as one figure. And the image resolution of the figures should be improved.

After the above problems have been carefully addressed, the manuscript is suitable for publication in Royal Society Open Science.

Review form: Reviewer 2

Is the manuscript scientifically sound in its present form?

Yes

Are the interpretations and conclusions justified by the results?

Yes

Is the language acceptable?

Yes

Is it clear how to access all supporting data?

Not Applicable

Do you have any ethical concerns with this paper?

No

Have you any concerns about statistical analyses in this paper?

No

Recommendation?

Accept with minor revision (please list in comments)

Comments to the Author(s)

The review titled "The application of metal-organic frameworks in the electrode materials for lithium-ion batteries and lithium sulfur batteries" by Zhu et al. covers the literature of some of the hottest aspects in the field of electrochemical energy storage. Most of the previous comments

have been addressed with care. The manuscript looks well-organized and of high technical standard. Hence, it can be accepted after following minor revisions.

1. The role of MOFs in MOFs based composites should be clearly highlighted w.r.t. electrochemical energy storage.
2. Why MOF in its original (pristine) form do not perform well for batteries system?
3. Following MOF based articles on other electrochemical energy storage systems can be cited in the introduction: J. Mater. Chem. A, 2016,4, 16432-16445; J. Mater. Chem. A, 2017,5, 17998-18011; J. Mater. Chem. A, 2019,7, 1725-1736.

Decision letter (RSOS-190634.R0)

28-May-2019

Dear Professor Zhu:

Title: The application of metal-organic frameworks in the electrode materials for lithium-ion batteries and lithium-sulfur batteries
Manuscript ID: RSOS-190634

Thank you for submitting the above manuscript to Royal Society Open Science. On behalf of the Editors and the Royal Society of Chemistry, I am pleased to inform you that your manuscript will be accepted for publication in Royal Society Open Science subject to minor revision in accordance with the referee suggestions. Please find the reviewers' comments at the end of this email.

The reviewers and handling editors have recommended publication, but also suggest some minor revisions to your manuscript. Therefore, I invite you to respond to the comments and revise your manuscript.

Please also include the following statements alongside the other end statements. As we cannot publish your manuscript without these end statements included, if you feel that a given heading is not relevant to your paper, please nevertheless include the heading and explicitly state that it is not relevant to your work. We have included a screenshot example of the end statements for reference.

- Funding statement

Please include a funding section after your main text which lists the source of funding for each author.

Because the schedule for publication is very tight, it is a condition of publication that you submit the revised version of your manuscript before 06-Jun-2019. Please note that the revision deadline will expire at 00.00am on this date. If you do not think you will be able to meet this date please let me know immediately.

When submitting your revised manuscript, you will be able to respond to the comments made by

the referees and upload a file "Response to Referees" in "Section 6 - File Upload". You can use this to document any changes you make to the original manuscript. In order to expedite the processing of the revised manuscript, please be as specific as possible in your response to the referees.

Best wishes,
Dr Laura Smith
Publishing Editor, Journals

RSC Associate Editor:
Comments to the Author:
(There are no comments.)

RSC Subject Editor:
Comments to the Author:
(There are no comments.)

Reviewer comments to Author:
Reviewer: 1

Comments to the Author(s)

In this manuscript, the authors summarized the research and application of MOFs in the area of lithium-ion batteries and lithium-sulfur batteries. The content of the manuscript is of logic and significance. But there are some problems as follows:

(1) The authors wrote that “Zhou et al [101] proposed a compartment with MOFs, which plays an important role in ion separation in lithium-sulfur batteries. Its working principle is shown in Figure 31.” But I don’t think this part belongs to MOF-derived carbon electrode materials for lithium-sulfur batteries. It should be MOF-based separator for lithium-sulfur batteries. The authors can deal with them separately and add some similar content and tables for better comparison.

(2) Many figures are chosen with no logic. The structure and electrochemical performance should be shown as one figure. And the image resolution of the figures should be improved.

After the above problems have been carefully addressed, the manuscript is suitable for publication in Royal Society Open Science.

Reviewer: 2

Comments to the Author(s)

The review titled “The application of metal-organic frameworks in the electrode materials for lithium-ion batteries and lithium sulfur batteries” by Zhu et al. covers the literature of some of the hottest aspects in the field of electrochemical energy storage. Most of the previous comments have been addressed with care. The manuscript looks well-organized and of high technical standard. Hence, it can be accepted after following minor revisions.

1. The role of MOFs in MOFs based composites should be clearly highlighted w.r.t. electrochemical energy storage.

2. Why MOF in its original (pristine) form do not perform well for batteries system?

3. Following MOF based articles on other electrochemical energy storage systems can be cited in the introduction: J. Mater. Chem. A, 2016,4, 16432-16445; J. Mater. Chem. A, 2017,5, 17998-18011; J. Mater. Chem. A, 2019,7, 1725-1736.

Author's Response to Decision Letter for (RSOS-190634.R0)

See Appendix A.

RSOS-190634.R1 (Revision)

Review form: Reviewer 1

Is the manuscript scientifically sound in its present form?

Yes

Are the interpretations and conclusions justified by the results?

Yes

Is the language acceptable?

Yes

Have you any concerns about statistical analyses in this paper?

No

Recommendation?

Accept as is

Comments to the Author(s)

The problems have been carefully addressed. The manuscript is suitable for publication in Royal Society Open Science.

Review form: Reviewer 2

Is the manuscript scientifically sound in its present form?

Yes

Are the interpretations and conclusions justified by the results?

Yes

Is the language acceptable?

Yes

Do you have any ethical concerns with this paper?

No

Recommendation?

Accept as is

Comments to the Author(s)

The revised manuscript can be accepted.

Decision letter (RSOS-190634.R1)

11-Jun-2019

Dear Professor Zhu:

Title: The application of metal-organic frameworks in the electrode materials for lithium-ion batteries and lithium-sulfur batteries

Manuscript ID: RSOS-190634.R1

It is a pleasure to accept your manuscript in its current form for publication in Royal Society Open Science. The chemistry content of Royal Society Open Science is published in collaboration with the Royal Society of Chemistry.

RSC Associate Editor:
Comments to the Author:
(There are no comments.)

RSC Subject Editor:
Comments to the Author:
(There are no comments.)

Reviewer(s)' Comments to Author:
Reviewer: 2
Comments to the Author(s)
The revised manuscript can be accepted.

Reviewer: 1
Comments to the Author(s)
The problems have been carefully addressed. The manuscript is suitable for publication in Royal Society Open Science.

Appendix A

Response to reviewers

We would like to thank the editor and the reviewers for the valuable comments and good suggestions. We have incorporated most of the reviewers' comments and suggestions into the revised manuscript, and also provided more explanations to the reviewers' questions.

Reviewer(s)' Comments:

Referee: 1

Comments:

In this manuscript, the authors summarized the research and application of MOFs in the area of lithium-ion batteries and lithium-sulfur batteries. The content of the manuscript is of logic and significance. But there are some problems as follows:

1. The authors wrote that “Zhou et al [101] proposed a compartment with MOFs, which plays an important role in ion separation in lithium-sulfur batteries. Its working principle is shown in Figure 31.” But I don't think this part belongs to MOF-derived carbon electrode materials for lithium-sulfur batteries. It should be MOF-based separator for lithium-sulfur batteries. The authors can deal with them separately and add some similar content and tables for better comparison.

In the revised manuscript, we delete this part that does not belong to electrode materials, because we mainly introduce electrode materials of lithium-ion batteries and lithium-sulfur batteries in our manuscript, this part belongs to the separator of batteries and does not belong to the electrode materials, so we delete this part from initial manuscript.

2. Many figures are chosen with no logic. The structure and electrochemical performance should be shown as one figure. And the image resolution of the figures should be improved.

The result of the modification is to make the selection of pictures more logical and to combine structure and performance on a single figure. The resolution of many images has been improved as much as possible.

Referee: 2

Comments:

Comments to the Author(s)

The review titled “The application of metal-organic frameworks in the electrode materials for lithium-ion batteries and lithium sulfur batteries” by Zhu et al. covers the literature of some of the hottest aspects in the field of electrochemical energy storage. Most of the previous comments have been addressed with care. The manuscript looks well-organized and of high technical standard. Hence, it can be accepted after following minor revisions.

1. The role of MOFs in MOFs based composites should be clearly highlighted w.r.t. electrochemical energy storage.

In the revised manuscript, we added a paragraph to emphasize the role of MOFs in MOFs-based composites after introducing lithium-sulfur batteries. In the last part, we emphasize the role of MOFs in MOF-based composites by introducing composite electrodes.

2. Why MOF in its original (pristine) form do not perform well for batteries system?

Conductivity and stability are the main obstacles to the application of MOFs in batteries systems. Many of the original forms of MOF are unstable in the air and can only be preserved in specific solvents, normally, the conductivity of original form of MOF is not very good, so MOF in its original (pristine) form do not perform well for batteries system. In the revised manuscript, at the beginning of the MOF-derived electrode materials for lithium-ion batteries and the end of MOF-derived carbon electrode materials for lithium-sulfur batteries, we have a brief introduction of the additions.

3. Following MOF based articles on other electrochemical energy storage systems can be cited in the introduction: J. Mater. Chem. A, 2016,4, 16432-16445; J. Mater. Chem. A, 2017,5, 17998-18011; J. Mater. Chem. A, 2019,7, 1725-1736.

We quote some of the articles mentioned by the reviewer. References 110, 111, 112 can be found in the revised manuscript.